# A Study of Metabolites from Basidiomycota and Their Activities against *Pseudomonas aeruginosa*

**DOI:** 10.3390/antibiotics13040326

**Published:** 2024-04-03

**Authors:** Marco Clericuzio, Giorgia Novello, Mattia Bivona, Elisa Gamalero, Elisa Bona, Alice Caramaschi, Nadia Massa, Alberto Asteggiano, Claudio Medana

**Affiliations:** 1Dipartimento di Scienze e Innovazione Tecnologica, Università del Piemonte Orientale, Viale T. Michel 11, 15121 Alessandria, Italy; marco.clericuzio@uniupo.it (M.C.); giorgia.novello@uniupo.it (G.N.); mattia.bivona@uniupo.it (M.B.); elisa.gamalero@uniupo.it (E.G.); 2Dipartimento per lo Sviluppo Sostenibile e la Transizione Ecologica, Università del Piemonte Orientale, Piazza San Eusebio 5, 13100 Vercelli, Italy; alice.caramaschi@uniupo.it; 3Struttura Semplice Dipartimentale Laboratori di Ricerca—Dipartimento Attività Integrate Ricerca e Innovazione, Azienda Ospedaliera SS. Antonio e Biagio e Cesare Arrigo, 15121 Alessandria, Italy; 4Dipartimento di Biotecnologie Molecolari e Scienze per la Salute, Università di Torino, Via Pietro Giuria 5, 10125 Torino, Italy; alberto.asteggiano@unito.it (A.A.); claudio.medana@unito.it (C.M.)

**Keywords:** *Pseudomonas aeruginosa*, fungal extract, HPLC-HRMS, NMR

## Abstract

The World Health Organization (WHO) promotes research aimed at developing new drugs from natural compounds. Fungi are important producers of bioactive molecules, and they are often effective against other fungi and/or bacteria and are thus a potential source of new antibiotics. Basidiomycota crude extracts, which have previously been proven to be active against *Pseudomonas aeruginosa* ATCC27853, were subjected to liquid chromatographic separation by RP-18, leading to six macro-fractions for each fungal extract. The various fractions were tested for their bioactivities against *P. aeruginosa* ATCC27853, and ten of them were characterized by HPLC-HRMS and NMR. Further chromatographic separations were performed for a few selected macro-fractions, yielding seven pure compounds. Bioactivity was mainly found in the lipophilic fractions containing fatty acids and their derivatives, such as hydroxy or keto C-18 unsaturated acids, and in various complex lipids, such as glycolipids and related compounds. More hydrophilic molecules, such as GABA, phenethylamine, two chromogenic anthraquinoids and pistillarin, were also isolated, and their antibacterial activities were recorded. The novelties of this research are as follows: (i) the genera *Cortinarius* and *Mycena* have never been investigated before for the synthesis of antibiotic compounds; (ii) the molecules produced by these genera are known, but their production has never been reported in the investigated fungi; (iii) the determination of bacterial siderophore synthesis inhibition by certain compounds from *Cortinarius* and *Mycena*.

## 1. Introduction

Antibiotic pressure has been established in recent years by drug misuses and bacteria reacted through different strategies, including molecule inactivation/modification, binding-site/target modifications and alterations in the cell permeability [1]. All these survival strategies lead to antimicrobial resistance (AMR), which is currently a worldwide public health emergency. In fact, AMR is expensive considering both the human health and social costs. According to the last report of the Organization for Economic Cooperation and Development (OECD) released in 2018, antidrug-resistant bacteria cause about 60,000 deaths each year, of which 33,000 occur in EU/EEA countries, and 29,500 occur in the United States. The projection model estimates that by 2050, 2.4 million people will have died due to infections carried out by antibiotic-resistant bacteria, with the costs reaching around USD 3.5 billion per year. In order to face AMR, a Global Action Plan was released in 2015 by the World Health Organization (WHO) [2].

In more detail, bacterial pathogens that cause hospital-acquired infections have been grouped by the Infectious Diseases Society of America under the acronym “ESKAPE”, which brings together the bacterial species *Enterococcus faecium, Staphylococcus aureus, Klebsiella pneumoniae, Acinetobacter baumannii, Pseudomonas aeruginosa* and *Enterobacter* spp. The main common characteristic among these bacterial species is their ability to escape from the biocidal activity of antibacterial drugs. Even more, ESKAPE bacteria frequently show a multidrug-resistant phenotype, which represents a serious risk, especially for immunocompromised patients. Based on this premise, ESKAPE bacterial pathogens have been included by the World Health Organization (WHO) among the twelve bacterial species for which the quick development of new antibiotic molecules is required (https://www.who.int/news/item/27-02-2017-who-publishes-list-of-bacteria-for-which-new-antibiotics-are-urgently-needed, accessed on 20 March 2024). In particular, the identification of innovative therapeutic compounds for *P. aeruginosa*, which usually behaves as an opportunistic pathogen and causes urinary, respiratory and bloodstream infections resistant to carbapenem, is defined as a critical priority. In addition to multidrug resistance, *P. aeruginosa* is equipped with an arsenal of highly destructive virulence factors (pili, lipopolysaccharide, elastase, protease, exotoxins, siderophores and so on) (for a recent review, see Liao et al., 2022) [3]. Among them, pyocyanin is a blue redox-active secondary metabolite that is often found in large amounts in the sputum from patients affected by cystic fibrosis. Once it crosses the cytoplasmic membrane, the oxidative stress induced by pyocyanin leads to cytotoxicity in the host cells [4], which is expressed by increased levels of Reactive Oxygen Species (ROS), the enhanced redox potential of cytosol and reduced ATP synthesis. Moreover, alterations in lymphocyte proliferation, macrophage function and ciliary beating and the increased secretion of mucous related to pyocyanin have been reported in the literature [4]. 

In this scenario, the search for new natural compounds that demonstrate bacterial-growth-inhibiting activity is of particular interest. Fungi [5] and plants [6,7,8,9] represent an important reservoir of bioactive substances that can serve, after appropriate characterization, as new antibacterial agents, either alone or in combination with drugs already in use. In particular, fungi are known to produce antibacterial metabolites; thus, they offer new promising antibiotic compounds [10]. Various screenings of basidiomes for antibiotic activity have recently appeared in the literature [11,12,13,14,15], and several extracts have shown significant results. In a previously published paper, the authors investigated the antibiotic activity of some extracts from the fruiting bodies of Agaricomycetes collected in the wild [5]. Some of them induced a significant inhibition of the *P. aeruginosa* growth and/or interference with pyocyanin synthesis. In the present paper, some fractions obtained after the liquid chromatographic separation of some of the abovementioned extracts were tested for their biological activities (19 for the growth inhibition assays and 16 for the siderophore production inhibition assays). In a few cases, a single molecule was obtained and tested for its bioactivity.

## 2. Results and Discussion

### 2.1. Biological Activities of Fractions

The biological activities of the fractions from the crude extracts of the different fungal species are reported in Figure 1 (*P. aeruginosa* ATCC27853 growth inhibition) and Figure 2 (inhibition of the siderophore production). Fractions inducing inhibition halo values similar to or higher than those of Imipenem (A) and Meropenem (B) were considered effective against *P. aeruginosa*, as underlined in the figure by the two blue lines.

Based on this interpretation, the effect of the fractions 2E, 9E, 15E, 15F, 16A and 16B were like those of the tested antibiotic drugs, while the fractions 2F, 9F, 16E and 16F performed better than the considered antibiotics (+35.1%, +25.2%, +12.6% and +51.4% versus Meropenem, respectively).

Interestingly, the fractions 1B/5 and 1B/12 did not induce a growth inhibition, but they had a high negative effect on the siderophore production, while the fractions 9E and 16E showed both biological activities. Fraction 16F strongly inhibited growth but stimulated siderophore production.

### 2.2. Chromatographic Separation and Chemical Characterization of Active Fractions

The raw extracts of *Cortinarius mussivus*, *C. caesiocanescens*, *Mycena renati* e *M. zephyrus* and *Ramaria parabotrytis*, obtained as reported by Clericuzio and coworkers [5], were submitted to LC RP-18 separation, according to a stepwise gradient. Other extracts treated in our previous investigation were discarded owing to the material paucity. After the preparative chromatography separation, each fraction was analyzed by HPLC-HRMS and NMR.

The results obtained for the fractions from the crude extracts from each fungal species are presented and discussed below. For coherence with our previous work, we here retained the same numbers of the fungal species as in Clericuzio et al. [5].

#### 2.2.1. *Cortinarius mussivus* (Species 1)

*C. mussivus* is a fungal species belonging to section *Percomes* of the *Cortinarius* subgenus *Phlegmacium*, the fruiting bodies of which are known to synthesize anthraquinones and dehydroanthraquinones, often in a dimeric form [16]. Anthraquinoides from Basidiomycota (mainly isolated from the genera *Cortinarius* and *Tricholoma*) are yellow–orange pigments with antibacterial and antifungal activities [17]. For instance, three monomeric tetrahydroanthraquinones isolated from *Cortinarius (Dermocybe) splendidus* were active against *Bacillus brevis*, *B. subtilis*, *Mucor miehei*, *Penicillium notatum* and *Nematospora coryli* [18].

A problem arising from this class of molecules is their marked lability: they are degraded in a few days, even at −20 °C. For this reason, the macro-fractions 1A–F obtained from *C. mussivus* were not subjected to the biological tests. Fraction 1B was the richest in phenolic pigments, judging from the TLC and NMR data; hence, we proceeded to its further fractioning. By means of direct-phase liquid chromatography on silica gel, we obtained, among others, subfractions 1B/5 and 1B/12. The most abundant compound in fraction 1B/5 was dehydrophlegmacin-9,10-quinone-8′-methylether **1a** (Figure 3). Similarly, dehydrophlegmacin-9,10-quinone **1b** (Figure 3) was the most abundant compound in fraction 1B/12. Compound **1a** (Table 1, Appendix A)) has already been isolated from *C. mussivus* (reported as the synonym *C. russeoides*) [16] and from other *Percomes* species, particularly *C. percomis* [19] and *C. nanceiensis* [16].

Fractions 1B/5-6 and 1B/12-15 did not inhibit *P. aeruginosa* growth but lowered the amount of siderophore released by the bacterial strain.

#### 2.2.2. *Cortinarius caesiocanescens* (Species 2)

Bioactive fractions 2E and 2F, showing a high rate of *P. aeruginosa* growth inhibition, were submitted to chemical characterization. The LC-MS chromatogram of fraction 2E (ESI+) is reported in Figure 4.

NMR analysis of fractions 2E and 2F showed similar spectra, revealing the dominance of fatty acid derivatives. Consequently, fractions 2E and 2F were mixed and submitted to direct-phase LC, yielding 25 fractions. An NMR analysis of these fractions showed the presence of diglycerides in the less polar fractions and complex lipids in the more polar ones. These fractions were not purified or characterized any further.

HPLC-HR-MS analyses confirmed the NMR results via the annotation of the lipidic structures, such as phosphocholine C18 isomers and oxo-octadecadienoic acid isomers (Table 2). Moreover, a bioactive metabolite from the plant *Orthosiphon aristatus* has been reported. In particular, its growth inhibition activity against *P. aeruginosa* and *Escherichia coli* is reported by Wahab and Chua [20].

Fraction 2E-F/20-21, for which the NMR suggested a complex lipidic structure, showed a low siderophore inhibition degree (Figure 2).

#### 2.2.3. *Cortinarius variicolor* (Species 3)

The raw extract of this fungus was fractionated, although it was not active in our past analysis [5]. A few pure compounds could be isolated. From macro-fraction 3A, the mammal neurotransmitter γ-aminobutirric acid (GABA) was isolated. This molecule has been sparsely reported as produced from fungi and plants [21,22,23]. From macro-fraction 3B, 3-hydroxybenzilic alcohol **2a** and 3-hydroxybenzoic acid **2b** were isolated (Figure 3, Appendix A).

The presence of such simple phenolics is interesting, and we could not find any past report of their occurrence in Basidiomycota.

#### 2.2.4. *Ramaria parabotrytits* (Species 9)

Active macro-fractions 9A, 9E and 9F were submitted to chemical analysis.

The metabolite **3** (Table 3) was isolated from the hydrophilic fraction 9A, showing mass peaks at 418.1967 (ESI+) and 416.1808 (ESI-). These peaks were consistent with a molecular mass of 417 and a C_21_H_27_N_3_O_6_ molecular formula. After extensive NMR investigation, and comparison with the literature data, we identified this compound **3** as pistillarin, a bis-(3,4-dihydroxybenzoic acid) amide derived from spermidine (Figure 3, Table 3, Appendix A). Pistillarin was initially isolated from *Clavariadelphus pistillaris*, and subsequently from *Ramaria flava*, *R. formosa* and *R. mairei* [24]. These species are phylogenetically closely related to *Ramaria parabotrytis*. Compound **3** is a bitter-tasting substance, and it is reported to have antimalarial activity [25]. However, fraction 9A was found to be non-active toward *Pseudomonas aeruginosa* (Figure 1).

As concerns more lipophilic fractions, the NMR analysis of fraction 9E showed the presence of linolenic acid (mass peak at 279.2318 at ESI-) and peaks at 295.2267 *m/z* (Table 4).

The peak at 295.2267 *m/z* in the negative-ion mode and the peak at 297.2423 *m/z* in the positive-ion mode, corresponding to a MW of 296 and the molecular formula C_18_H_32_O_3_, were attributed, after 1- and 2-D NMR analysis, to 13-hydroxy-9Z,11E-octadecadienoic acid **4** (Table 4). This oxylipin, known as coriolic acid, is mainly present in nature as the 13*S* isomer, though, in our case, the absolute configuration could not be established (Figure 3). Coriolic acid is widespread both in plants and fungi and is reported to have several biological effects, including antibiotic, antioxidant, anti-inflammatory [26] and anticancer activities. In particular, regarding its antibiotic effect, using an agar-plate diffusion test, it was demonstrated that coriolic acid extracted from the cyanobacterium *Oscillatoria redekei* inhibited the growth of the Gram-positive bacteria *Bacillus subtilis* SBUG 14, *Micrococcus flavus* SBUG 16 and *Staphylococcus aureus* SBUG 11 and ATCC 25923 [27]. Concerning other biological activities, we can mention that (i) coriolic acid in the methanol extract from the plant *Chromolaena odorata* showed anti-inflammatory activity in murine cells [26]; (ii) (S)-coriolic acid extracted from *Salicornia herbacea* L. suppressed breast cancer stem cells through the regulation of c-MycCancer [28].

In addition, in the ^1^H NMR spectrum of fraction 9E, the presence of a fatty acid with a conjugated ketone group was visible (Appendix A). This finding was confirmed by MS analysis. In the chromatogram of Figure 5, three peaks at *m/z* 295.2267 (negative-ion mode) are visible, with R_T_ 23.3, 24.5 and 25.9 min, and their exact masses correspond to oxo octadecadienoic acids (Table 4). One of these three isomers is very likely 13-oxo-octadeca-9(Z),11(E)-dienoic acid (13-ketooctadienoic acid, 13-KODE) **5** (Figure 3, Appendix A), the oxidation product of coriolic acid. Compound **5** has been isolated from wounded and diseased *Arabidopsis* leaves [29], together with its isomer 9-KODE. This literature finding is suggestive that a second isomeric ketoacid in fraction 9E may be 9-KODE. Moreover, 9-KODE is reported as a bioactive metabolite from the plant *Orthosiphon aristatus*. In particular, its growth inhibition activity against *P. aeruginosa* and *Escherichia coli* is reported by Wahab and Chua [20].

#### 2.2.5. *Mycena renati* (15)

The six fractions 15A–F, obtained by the usual RP-18 separation of the crude extract, were obtained in very small amounts. NMR analysis of some selected fractions revealed the presence of widespread fungal metabolites, such as oleic acid, ergosterol peroxide and triglycerides.

HPLC-HRMS analysis was performed on the active fraction 15E (Table 5). The LC-MS ESI+ chromatogram of 15E is reported in Figure 6.

Fraction 15E resulted in the enrichment of lipophilic sphingosines, such as phytosphingosine and saginfol. These sphingosines are widely spread in the plant kingdom, as well as in fungi [30].

#### 2.2.6. *Mycena zephyrus* (16)

The most hydrophilic fraction, 16A, showed the chromatogram reported in Figure 7 (ESI+). From preparative HPLC RP-C18 separations, we could isolate phenethyl amine (PEA), a mammal neurotransmitter, sometimes reported in fungi [31]. In the chromatogram of Figure 7, its peak is the one at Rt 8 min, where *m/z* 105.0689 corresponds to M-NH_3_^+^ (Table 6).

The in-source loss of the NH_3_ hypothesis was confirmed by the direct injection into the HRMS instrument of purified PEA, where a less intense mass peak at *m/z* 122 was also visible.

This hydrophilic fraction showed, in addition to phenethyl amine, a few other molecules containing the same moiety, such as 4-[[(1S)-1-phenylethyl] amino] hexan-2-one and N-(2-Phenylethyl) acetamide, in addition to short-chain heptanamide and long-chain (N-Pentacosa-10,12-diynoylglycine) fatty acid amide.

Fraction 16B, the HPLC-HRMS chromatogram of which is reported in Figure 8 (ESI+), shows mostly low-molecular-weight signals.

Apart from azelaic and suberic acid methyl esters, this fraction is mostly characterized by the presence of nitrogen-containing compounds (Table 7). The mass library indicates the presence of phenilpropenilamino piperidinic acids. However, more investigations are needed to clarify the proper structure of this compound.

The presence of nitrogen-containing compounds in *M. zephirus* is a quite significant finding. In fact, it has long been known that some *Mycena* species have a hallucinogenic character, in particular *M. pura* and related taxa [32]. Consequently, all *Mycena* species have been regarded as non-edible or even toxic. The presence of muscarine and its isomers in *M. pura* has been proven by Stadelmann (1976) [33], who reported the dominant presence of epi-muscarine over muscarine (about 80:20). *Mycena zephirus* belongs to a different section than *M. pura* (*Fragilipedes* vs. *Calodontes*) [34], which means that the presence of alkaloids in this genus is not restricted to the *M. pura* complex. Moreover, the recent finding of several pyrroloquinoline alkaloids [35,36,37,38,39] in *M. rosea* (section *Calodontes*), *M. haematopus* (*Galactopoda*) and *M. sanguinolenta* (*Sanguinolentae*) fully confirms the widespread presence of nitrogen-containing metabolites in the genus. Because *Mycena* is a very large cosmopolitan genus with more than one thousand species worldwide, this group of fungi has significant potential as a source of bioactive metabolites.

The LC-MS chromatograms of the lipophylic 16E fraction are shown in Figure 9 (ESI+). The presence of linolenic, linoleic and palmitic acid was confirmed by both HPLC and NMR analyses (Table 8).

Similar to the *Cortinarius caesiocanescens* and *Ramaria parabotrytits* lipophilic fractions, HPLC-HRMS analyses confirmed the presence of LPCs, in addition to other C18- and C21-N-containing fatty acids.

The 16F macro-fraction was furtherly separated by LH-20 size-exclusion chromatography, yielding 20 fractions. Fractions 16F/2 and 16F/15 showed similar ^1^H NMR spectra (Appendix A), in which a complex lipidic structure is highlighted. In particular, signals for long aliphatic chains and broad signals in the oxygenated carbon region (4.6–3.2 ppm) were observed. The MS of this fraction confirmed the presence of high-MW molecules.

## 3. Materials and Methods

### 3.1. Reagents

Acetonitrile (ACN) and methanol (MeOH) (Chromanorm) and the formic acid Emparta (FA) were purchased from VWR (Milan, Italy). Ultrapure water was obtained via a Milli-Q apparatus by the Millipore corporation (Burlington, MA, USA).

### 3.2. Analytical Sample Preparation

Powder samples coming from the fractionation, prior to the HPLC-HRMS and Flow Injection analysis coupled with High-Resolution Mass Spectrometry (FIA-HRMS) analysis, were resuspended in ACN. After the complete solubilization, the extract was diluted 100-fold in the solvent of the initial conditions of the HPLC run.

### 3.3. FIA and HPLC-HRMS Analysis

The analytical setup consisted of an Orbitrap Fusion trihybrid (Thermo Scientific, Palo Alto, Santa Clara, CA, USA) coupled with an LPG DIONEX 3000 HPLC (Thermo Scientific, Palo Alto, Santa Clara, CA, USA) H-ESI ion source. FIA analysis was performed by the direct injection of the liquid samples into the H-ESI source using a 4.27 mm i.d. glass syringe (500 μL) and pumping at 10 μL/min. The HPLC run consisted of a binary gradient of A: Water 0.1 AF and B: ACN 0.1% AF. The chosen column was a Luna C18 (2 × 150 mm, 3 μm), and the gradient was set as follows: flow: 0.2 mL/min; initial conditions: t = 0 min, B% = 5; t = 4 min, B% = 5; t = 35 min, B% = 100; t = 37 min, B% = 100; t = 44 min, B% = 5, and the reconditioning step was carried out till the 50th minute. The column oven was set to maintain a temperature of 40 °C. HRMS was operated in both ionization modes, the sheath gas was 35 arb, the auxiliary gas was 20 arb, the capillary temp. was 275 °C and the interface voltage was 4500 V in the positive-ion mode and 3800 V in the negative-ion mode. The experiments were conducted in DDA, the surveyor event was a full scan in the range between 100 and 1000 *m/z,* and the resolution was set at 50 000, the intensity trigger for the MS2 was set at 10,000 and the CID collision energy was 25.

### 3.4. HPLC-HRMS Data Analysis

Raw data obtained by HPLC-HRMS runs were submitted to a feature detection and annotation pipeline consisting of raw data conversion with *MsConvert* (Proteowizard, Palo Alto, Santa Clara, CA, USA) in MzXmL centroided data. After the conversion, peak picking and feature detection were performed with MzMine 3.9.0. Lastly, the feature annotation step was conducted with Sirius 5.8.5 (Friedrich-Schiller-Universität Jena, Jena, Germany) [40]. The annotations were manually confirmed. For all the compounds, the annotation level, according MSI, is the 2nd level [41].

### 3.5. Preparative Liquid Chromatography

Macro-fractions A–F were obtained by means of an MPLC apparatus consisting of an Alltech 426 HPLC pump equipped with a VWR (Milan, Italy) LaPrep 3101 UV detector. The column employed was a Merck Lichrosphere 100 RP-18e 250 × 22 mm, 20 μm. The elution gradient was that reported in Table 9. The flow rate was set at 12.0 mL/min.

Further fractioning was achieved by a Merck Lichrosorb (Darmstadt, Germany) RP-18 250 mm × 10 mm, 10 μm (reversed phase), or a Merck Lichrosorb Si 100 250 mm × 10 mm, 10 μm (direct phase).

### 3.6. NMR Spectroscopy

The NMR spectra were recorded on a Bruker Avance III 500 MHz spectrometer (Karlsruhe, Germany), operating at 499.802 MHz (^1^H) and 125.687 MHz (^13^C).

### 3.7. Antibacterial Assays and Siderophore Production Assay

The antibacterial and siderophore production method are fully described in Clericuzio et al. [5]. Briefly, the antibacterial assay was carried out using the strain *Pseudomonas aeruginosa* ATCC27853. The Imipenem and Meropenem effects were evaluated according to the EUCAST Disk Diffusion Method for Antimicrobial Susceptibility v. 14.0/2024 (January 2024). Strain suspensions (0.5 McFarland), obtained in physiological solution, were swabbed on Mueller Hinton Agar (Biolife Italiana s.r.l., Monza, Italy) plates. Filter-paper discs (6.0 mm diameter) were placed on the agar surface and 10 μL of extracts was added to evaluate their antibacterial activities. 1,4 Dioxane (Sigma-Aldrich, St. Louis, MO, USA, (10 μL) and organic linseed oil (10 μL) disks were used as negative controls. Plates were incubated at 37 °C for 24 h. All experiments were performed in triplicate. The halos were measured in millimeters using calipers. The extract was considered active when it produced a halo equal to or higher than that of the positive control (positive control ≥ 100%).

Siderophore production was evaluated on Chrome Azurol S (CAS) agar according to Schwyn and Neilands [42]. The bacterial strains were inoculated at the center of each plate and incubated at 28 °C for seven days. In order to evaluate the possible siderophore synthesis inhibition, a filter-paper disk (6 mm) was placed on the colony and 10 μL of extracts resuspended in dioxane was added. The ability to produce siderophore was indicated by the occurrence of a yellow–orange halo around the colony and was measured with calipers as the ratio between the two diameters of the halo and the two diameters of the colony.

### 3.8. Statistical Analysis

Data regarding biological activity were tested for normality and homogeneity of variances. Thereafter, parametric (ANOVA, Welch one-way ANOVA) or non-parametric (Kruskal–Wallis, permutation test) one-way tests were performed to compare each fraction to each control (Imipenem or Meropenem for the parameter Halo; Control or Dioxane for the parameter Inhibition). The statistical analysis was performed using R (v. 3.5.1) [43]. Data are presented as boxplots. Differences were considered significant for *p*-values < 0.05.

## 4. Conclusions

The fruiting bodies of Basidiomycota are a rich source of metabolites, and these are often different from those found in the mycelia.

Here, we demonstrated how a systematic search for antibiotics can allow us to find important bioactive compounds, often still undescribed. The large genus *Mycena* is still a new investigation field, as only a handful species have been investigated so far, despite the worldwide occurrence of more than one thousand species. *Cortinarius*, by far the most species-rich genus of Agaricomycetes (no less than 2500 species have been described to date), has been somehow more studied, mainly for toxins (e.g., orellanin) and pigments (antraquinoids), but the great majority of species are still uninvestigated.

Phytochemical investigations of fruiting bodies collected in the wild have several serious drawbacks. Large amounts of them (often in multi-kilo amounts) are needed, as many interesting metabolites are often present in tiny quantities. Moreover, fungal metabolites are often not stable, and they may degrade in a few weeks, even if kept at −20 °C. Also, incorrect extraction or separation methods may lead to the formation of artifacts.

HPLC-HRMS has proven to be a powerful analytical technique for the characterization of metabolites in such complex mixtures. In this study, the application of HPLC-HRMS provided valuable insights into the compositions of various fractions and species, offering a comprehensive profile of the metabolites present. The high resolution and accuracy of mass spectrometry allowed for the precise determination of the molecular formulae and their putative identification through structural elucidation. The best results are achieved when NMR data can be coupled with MS data, but this requires larger amounts of material and the possibility of obtaining pure compounds.

## Figures and Tables

**Figure 1 antibiotics-13-00326-f001:**
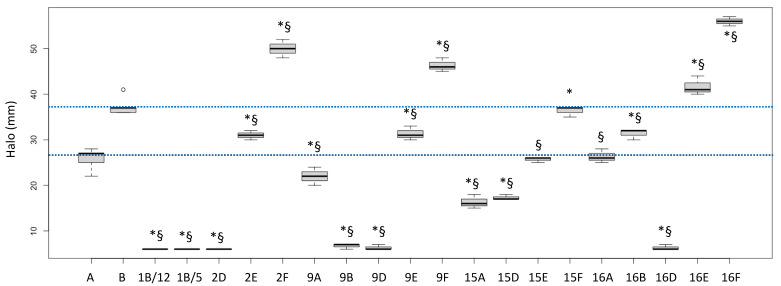
Halos (biological activities) induced by some fractions obtained from crude extracts of fruiting bodies of different fungi induced in *P. aeruginosa* ATCC27853. 1: *Cortinarius mussivus;* 2: *Cortinarius caesiocanescens;* 9: *Ramaria parabotrytits;* 15: *Mycena renati*; 16: *Mycena zephirus*. Halos were measured using calipers. A: Imipenem; B: Meropenem. * and § indicate significant differences in one-way tests (*p* < 0.05) between each fraction and Imipenem (A) or Meropenem (B), respectively. The two dotted lines indicate the range of activity of the two antibiotics used as positive controls.

**Figure 2 antibiotics-13-00326-f002:**
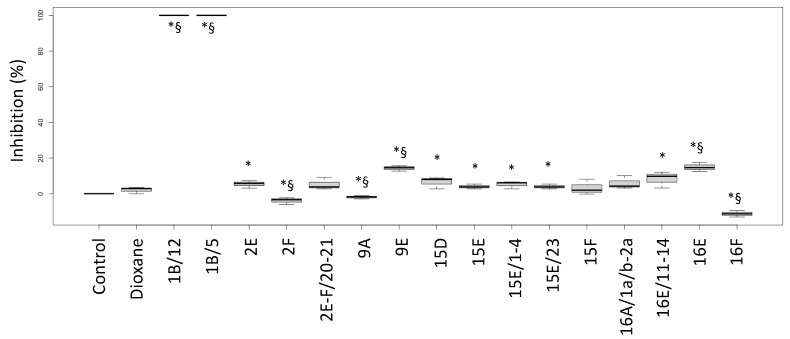
Inhibition percentages of siderophore production (biological activity) induced by fractions from the crude extracts obtained from fruiting bodies of different fungi induced in *P. aeruginosa* ATCC27853. Labels: Control: negative control (only medium); Dioxane: Dioxane (negative control); 1: *Cortinarius mussivus;* 2: *Cortinarius caesiocanescens*; 9: *Ramaria parabotrytits;* 15: *Mycena renati*; 16: *Mycena zephirus*. Halos were measured using calipers. * and § indicate significant differences in one-way tests (*p* < 0.05) between each fraction and Control or Dioxane, respectively.

**Figure 3 antibiotics-13-00326-f003:**
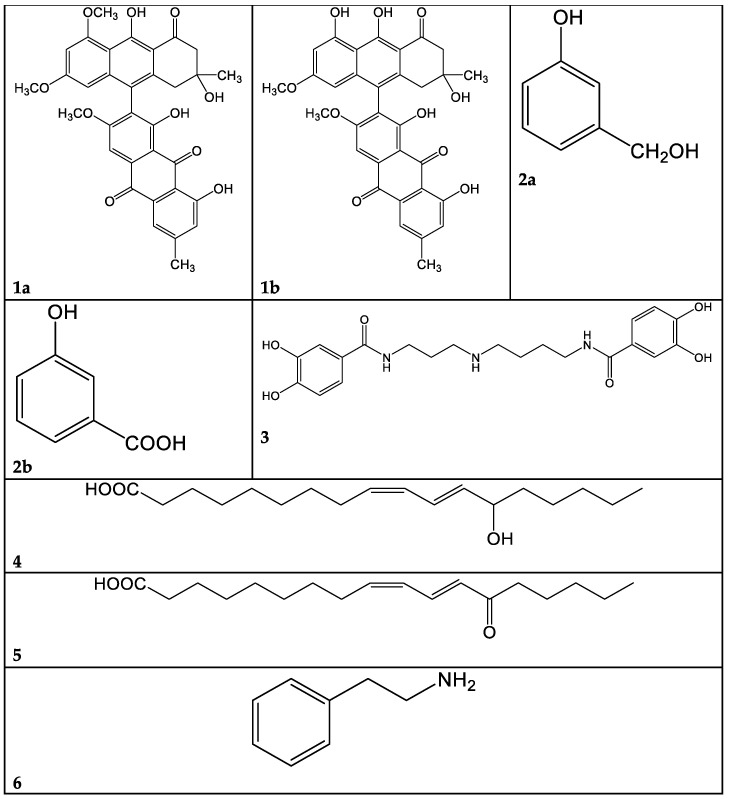
Selected compounds identified in the present work. **1a**: Dehydrophlegmacin-9,10-quinone-8′-methylether. **1b**: Dehydrophlegmacin-9,10-quinone. **2a**: 3-hydroxybenzilic alcohol. **2b**: 3-hydroxybenzoic acid. **3**: Pistillarin. **4**: Coriolic acid. **5**: 13-KODE. **6**: Phenethyl amine.

**Figure 4 antibiotics-13-00326-f004:**
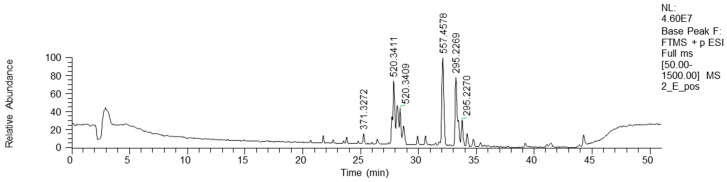
HPLC base peak ionic chromatogram of fraction 2E.

**Figure 5 antibiotics-13-00326-f005:**
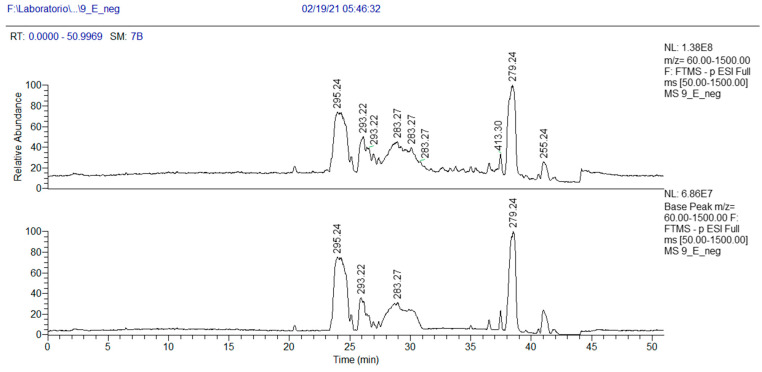
HPLC base peak ionic chromatogram of fraction 9E.

**Figure 6 antibiotics-13-00326-f006:**
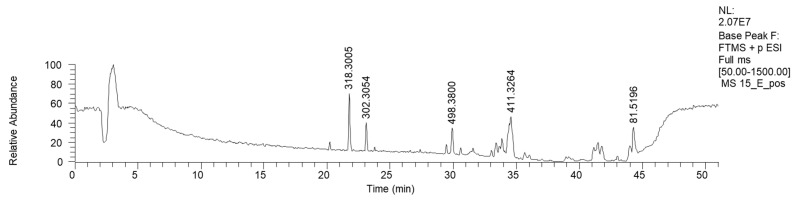
LC-HRMS base peak chromatogram of fraction 15E (ESI+).

**Figure 7 antibiotics-13-00326-f007:**
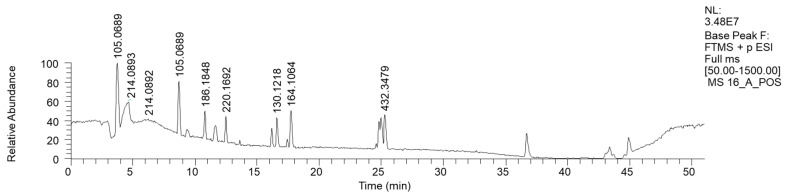
LC-HRMS base peak chromatogram of fraction 16A (ESI+). The peak with the *m/z* value of 105.0689 corresponds to M-NH_3_^+^ of PEA.

**Figure 8 antibiotics-13-00326-f008:**
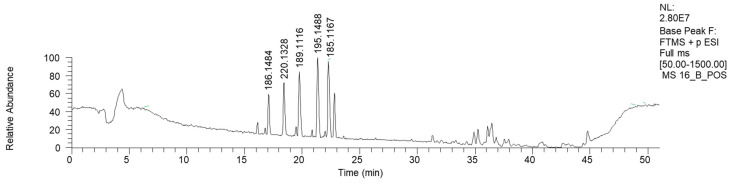
LC-HRMS base peak chromatogram of fraction 16B (ESI+).

**Figure 9 antibiotics-13-00326-f009:**
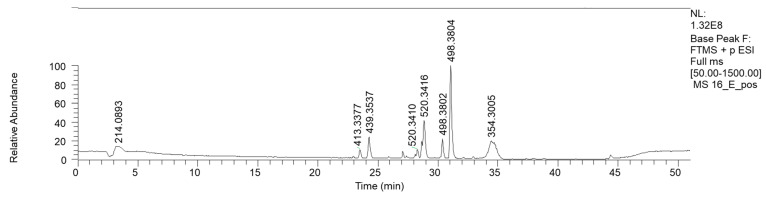
LC-HRMS base peak chromatogram of fraction 16E (ESI+).

**Table 1 antibiotics-13-00326-t001:** Compound **1a**: Calculated exact *m/z*, accurate *m/z*, Delta ppm and RDB (direct inlet).

Ion Formula	Exact *m/z*	Accurate *m/z*	Delta ppm	RDB
C_33_H_29_O_10_	+585.1760	+585.1764	1.27	19.5

**Table 2 antibiotics-13-00326-t002:** HRMS annotation of compounds in fractions 2E-F. For each putatively identified compound, the retention time (RT) (minutes), name, calculated formula, accurate mass, exact mass, delta ppm and fragment *m/z* values are reported.

RT	Putative Compound	Ion Formula	Accurate (+/−Polarity) *m/z*	Exact *m/z*	Delta ppm	Fragments
27.84	LPC 18:2 isomer 1	C_26_H_51_NO_7_P	(+) 520.3410	520.3421	−2.1	502.3308 (C_26_H_49_NO_6_P) 184.0724 (C_5_H_15_NO_4_P) 443.2571 (C_23_H_40_O_6_P)
28.61	LPC 18:2 isomer 2	C_26_H_51_NO_7_P	(+) 520.3416	520.3421	−1.0	502.3308 (C_26_H_49_NO_6_P) 184.0724 (C_5_H_15_NO_4_P) 443.2571 (C_23_H_40_O_6_P)
31.99	LPC 18:2 isomer 3	C_26_H_51_NO_7_P	(+) 520.3416	520.3421	−1.0	502.3308 (C_26_H_49_NO_6_P) 184.0724 (C_5_H_15_NO_4_P) 443.2571 (C_23_H_40_O_6_P)
33.11	13-Oxooctadecadienoic acid Isomer 1	C_18_H_31_O_3_	(+) 295.2269	295.2269	−0.01	277.2165 (C_18_H_29_O_2_) 221.1534 (C_14_H_21_O_2_)
34.0	13-Oxo-octadecadienoic acid Isomer 2	C_18_H_30_O_3_	(+) 295.2270	295.2269	−1.4	277.2165 (C_18_H_29_O_2_) 221.1534 (C_14_H_21_O_2_)

**Table 3 antibiotics-13-00326-t003:** Pistillarin (**3**): calculated exact *m/z*, accurate *m/z*, Delta ppm and RDB.

Proposed Name	Ion Formula	Exact *m/z*	Accurate *m/z*	Delta ppm	RDB
Pistillarin	+C_21_H_28_O_6_N_3_ −C_21_H_26_O_6_N_3_	+418.1945 −416.1789	+418.1966 −416.1809	+1.61 −2.26	10.5

**Table 4 antibiotics-13-00326-t004:** HRMS annotation of compounds in fraction 9E. For each putatively identified compound, the retention time (RT) (min), name, calculated formula, accurate mass, exact mass, delta ppm and fragment *m/z* values are reported.

RT	Putative Compound	Ion Formula	Accurate (+/−Polarity) *m/z*	Exact *m/z*	Delta ppm	Fragments
23.3	Oxo-octadecadienoic acid Isomer 1	C_18_H_31_O_3_	(+)295.2267	295.2269	0.43	277.2165 (C_18_H_29_O_2_) 221.1534 (C_14_H_21_O_2_)
24.5	Oxo-octadecadienoic acid Isomer 2	C_18_H_31_O_3_	(+) 295.2267	295.2269	0.43	277.2165 (C_18_H_29_O_2_) 221.1534 (C_14_H_21_O_2_)
25.9	Oxo-octadecadienoic acid Isomer 3	C_18_H_31_O_3_	(+) 295.2267	295.2269	0.43	277.2165 (C_18_H_29_O_2_) 221.1534 (C_14_H_21_O_2_)
27.8	LPC 18:2 isomer 1	C_26_H_51_NO_7_P	(+) 520.3397	520.3406	2.0	502.3308 (C_26_H_49_NO_6_P) 184.0724 (C_5_H_15_NO_4_P) 443.2571 (C_23_H_40_O_6_P)
28.3	LPC 18:2 isomer 2	C_26_H_51_NO_7_P	(+) 520.3397	520.3406	2.0	502.3308 (C_26_H_49_NO_6_P) 184.0724 (C_5_H_15_NO_4_P) 443.2571 (C_23_H_40_O_6_P)
32.0	Linolenic Acid	C_18_H_31_O_2_	(−) 279.2318	279.2320	0.37	261.2207 (C_18_H_29_O) 243.2110 (C_18_H_2_)
35.7	Coriolic acid	+C_18_H_33_O_3_ −C_18_H_31_O_3_	(+) 297.2424 (−) 295.2267	297.2425	0.23	279.2314 (C_18_H_31_O_2_) 183.1373 (C_11_H_19_O_2_)

**Table 5 antibiotics-13-00326-t005:** HRMS annotation of compounds in fraction 15E. For each putatively identified compound, the retention time (RT) (min), name, calculated formula, accurate mass, exact mass, delta ppm and fragment *m/z* values are reported.

RT	Putative Compound	Ion Formula	Accurate (+/−Polarity) *m/z*	Exact *m/z*	Delta ppm	Fragments
21.67	Phytosphingosine	C_18_H_40_NO_3_	(+) 318.3005	318.3003	0.721	300.2896 (C_18_H_40_NO_2_) 270.2790 (C_17_H_36_NO) 265.2525 (C_18_H_33_O)
23.18	Saginfol	C_18_H_40_NO_2_	(+) 302.3051	302.3054	−0.847	284.2944 (C_18_H_38_NO) 266.2843 (C_18_H_36_N) 240.2677 (C_16_H_34_N)

**Table 6 antibiotics-13-00326-t006:** HRMS annotation of compounds in fraction 16A. For each putatively identified compound, the retention time (RT) (min), name, calculated formula, accurate mass, exact mass, delta ppm and fragment *m/z* values are reported.

RT	Putative Compound	Ion Formula	Accurate (+/−Polarity) *m/z*	Exact *m/z*	Delta ppm	Fragments
8.01	Phenethyl amine	C_8_H_12_N	(+) 122.0956	122.0964	6.7	105.0690 (C_8_H_8_) 79.0352 (C_6_H_6_)
10.36	Propanamide, N, N-dibutyl-	C_11_H_24_NO	(+) 186.1848	186.1852	2.3	n.d.
12.29	4-[[(1S)-1-phenylethyl-amino hexan-2-one	C_14_H_22_NO	(+) 220.1692	220.1696	−1.7	162.1271 (C_11_H_15_N) 105.0691 (C_8_H_8_)
16.58	Heptanamide	C_7_H_16_NO	(+) 130.1218	130.1226	−6.5	60.0434 (C_7_H_15_NO)
17.51	N-(2-Phenylethyl) acetamide	C_10_H_14_NO	(+) 164.1064	164.1070	−3.6	105.0690 (C_8_H_8_)
25.1	N-Pentacosa-10,12-diynoylglycine	C_27_H_46_NO_3_	(+) 432.3479	432.3472	1.6	290.2121 (C_18_H_27_O_2_) 246.2216 (C_17_H_27_N) 232.2053 (C_16_H_25_N)

**Table 7 antibiotics-13-00326-t007:** HRMS annotation of compounds in fraction 16B. For each putatively identified compound, the retention time (RT) (min), name, calculated formula, accurate mass, exact mass, delta ppm and fragment *m/z* values are reported.

RT	Putative Compound	Formula	Accurate (+/−Polarity) *m/z*	Exact *m/z*	Delta ppm	Fragments
16.7	(3R)-3-[(2R)-piperidin-2-yl] pentanoic acid	C_10_H_20_NO_2_	(+) 186.1484	186.1489	−2.4	168.1379 (C_10_H_18_NO) 71.0846 (C_5_H_11_)
18.6	4-(1-Phenylprop-2-enylamino) butanoic acid	C_13_H_18_NO_2_	(+) 220.1328	220.1332	−1.8	122.0957 (C_8_H_12_N) 105.0689 (C_8_H_10_)
19.62	Monomethyl suberate	C_9_H_17_O_4_	(+) 189.1116	189.1121	−2.8	171.1006 (C_9_H_15_O_3_) 139.0746 (C_8_H_11_O_2_)
21.35	2-(2-Oxo-1-propan-2-yl piperidin-3-yl) propanenitrile	C_11_H_19_N_2_O	(+) 195.1488	195.1492	−2.0	125.1063 (C_7_H_13_N_2_) 180.1247 (C_10_H_16_N_2_O) 167.1538 (C_10_H_19_N_2_)
22.22	Monomethyl azelate	C_10_H_17_O_3_	(+) 185.1167	203.1278	0.01	153.0903 (C_9_H_13_O_2_) 125.0952 (C_8_H_13_O) 135.0797 (C_9_H_11_O) 107.0846 (C_8_H_11_)

**Table 8 antibiotics-13-00326-t008:** HRMS annotation of compounds in fraction 16E. For each putatively identified compound, the retention time (RT) (min), name, calculated formula, accurate mass, exact mass, delta ppm and fragment *m/z* values are reported.

RT	Putative Compound	Formula	Accurate (+/−Polarity) *m/z*	Exact *m/z*	Delta ppm	Fragments
23.5	Methyl 2,6-bis(octanoylamino)hexanoate	C_23_H_45_N_2_O_4_	(+) 413.3377	413.3373	−0.2	310.2739 (C_19_H_36_NO_2_) 176.0915 (C_7_H_14_NO_4_)
24.04	2-[[(Z)-13-[heptanoyl(propan-2-yl) amino] tridec-8-enoyl] amino]acetic acid	C_25_H_47_N_2_O_4_	(+) 439.3537	439.3530	1.5	379.3330 (C_23_H_43_N_2_O_2_) 338.3060 (C_21_H_34_NO_2_) 336.2890 (C_21_H_38_NO_2_)
27.84	LPC 18:2 isomer 1	C_26_H_51_NO_7_P	(+) 520.3410	520.3421	−2.1	502.3308 (C_26_H_49_NO_6_P) 184.0724 (C_5_H_15_NO_4_P) 443.2571 (C_23_H_40_O_6_P)
28.36	LPC 18:2 isomer 2	C_26_H_51_NO_7_P	(+) 520.3416	520.3421	−1.0	502.3308 (C_26_H_49_NO_6_P) 184.0724 (C_5_H_15_NO_4_P) 443.2571 (C_23_H_40_O_6_P)
34.3	2-(Linoleylamino)-1,3-propanediol	C_21_H_40_NO_3_	(+) 354.3005	354.3002	0.64	336.2892 (C_21_H_35_O_2_) 175.1475 (C_13_H_19_)

**Table 9 antibiotics-13-00326-t009:** Scheme of the eluents used to produce macro-fractions A–F from the Basidiomycota crude extracts.

Fraction Eluent	H_2_O	MeOH	Acetonitrile	Acetone
A	75	25	0	0
B	45	55	0	0
C	30	70	0	0
D	15	85	0	0
E	0	0	100	0
F	0	0	0	100

## Data Availability

Data are contained within the article and Appendix A.

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
