# Peer review of "A Study of Metabolites from Basidiomycota and Their Activities against Pseudomonas aeruginosa"

_antibiotics, 2024, doi:10.3390/antibiotics13040326_

Round 1

Reviewer 1 Report

Comments and Suggestions for Authors

The manuscript reports the isolation of some bioactive compounds from Basidiomycota extracts. Please carefully check the numbers of exact mass and accurate m/z in Table 1, as there is a significant difference: 575.1755 and 585.1764.

Essentially, ESI+ mass spectra give the molecular ion peak as [M+H]+. Therefore, the formula needs to include one proton when calculating the molecular mass. For example, the formula could be C33H28O10, not C33H27O10 for compound 1a in Table 1. The formulas in Tables 1 to 8 can be corrected accordingly. Similarly, if you use ESI- mass spectra, it should give the molecular ion peak as [M+H]-.

All the compounds isolated from Basidiomycota extracts have known chemical structures, but I don’t understand how their exact chemical structures were identified. Although the authors insist on "extensive NMR investigation and comparison with literature data," there is no such data provided in the manuscript. Therefore, I request that all data, including NMR spectra, be added to the supplementary information.

Additionally, the text in Figures 1 and 2 appears garbled, making it unreadable.

Author Response

Reviewer 1

The manuscript reports the isolation of some bioactive compounds from Basidiomycota extracts. Please carefully check the numbers of exact mass and accurate m/z in Table 1, as there is a significant difference: 575.1755 and 585.1764.

The second column describes the exact mass, while the second column the accurate mass over charge (m/z) and accounts the presence of a H+. The reviewer is right, there is a typo which has been corrected. For clarity, the table has been modified by substituting “exact mass” with Exact m/z in order to maintain the current article format. The Ionization polarity is reported.

Essentially, ESI+ mass spectra give the molecular ion peak as [M+H]+. Therefore, the formula needs to include one proton when calculating the molecular mass. For example, the formula could be C33H28O10, not C33H27O10 for compound 1a in Table 1. The formulas in Tables 1 to 8 can be corrected accordingly. Similarly, if you use ESI- mass spectra, it should give the molecular ion peak as [M+H]-.

Thank you for the suggestion, we modified the tables accordingly by using the term ION formula and not molecular formula.

All the compounds isolated from Basidiomycota extracts have known chemical structures, but I don’t understand how their exact chemical structures were identified. Although the authors insist on "extensive NMR investigation and comparison with literature data," there is no such data provided in the manuscript. Therefore, I request that all data, including NMR spectra, be added to the supplementary information.

A supplemental information file was produced with all NMR spectra (Figures S1-S16).

Additionally, the text in Figures 1 and 2 appears garbled, making it unreadable.

The Authors modified Figures 1 and 2 and they hope now they are easily readable.

Reviewer 2 Report

Comments and Suggestions for Authors

The authors should consider the followings:

The authors should state whether their identified peaks, even though with HR molecular formula estimation, did the authors verify those candidate peaks with purchased or in-house chemical markers for verification? What are the false discovery rate of all the candidates concern? Did the authors use other methodology to cross check the amount and validity of all the proposed candidates? If yes, please elaborate clearly and if no, the authors should justify why not.

How many of those candidate peaks verified by the chemical markers? And the authors should explain the ratio? 

"3.8. Statistical analysis Disk diffusion results were statistically analyzed by one-way ANOVA followed by Tukey HSD multiple comparisons of means using R (v. 3.5.1) [43]. Data are presented as boxplots. Differences were considered significant for p-values < 0.05.". The authors should make a better statisical analysis plan, and include the normality test, with pipeline for both parametric and non-parametric analyses. Sample size justification should also be discussed.

As for the Antibacterial assays, how did the authors verify the strain they used for testing? By pcr method?

"A filter paper disk 6 mm was placed on the colony and added with 10 μl of extracts or 10 μl of saline solution (0.8%) to evaluate the possible 

siderophore synthesis inhibition. The ability to produce siderophore was indicated by the occurrence of a yellow-orange halo around the colony..." The authors should explain whether the "extracts" contain interfering agents that affect the color test of the halo, which may create artifact results. Secondly, would the extract only dissolved in saline solution, or would the extract dissolved in other solvent? The authors should make sure that the solvent effect is well controlled in the experiments.

The authors should provide the chemical structutes of all the validated or proposed chemical candidates in a figure.

As in a supplementary figure, the authors may depict the mechanism of secondary metabolite(s) involved in the study may be illustrated. 

The authors may further give details whether any of the chemical candidates undergoing clinical trials by the team or others, and the results if the trials (microbiological specific).

Study limitations should be discussed in the article 

The authors should clearly state the novelty of the current study in the abstract.

Comments on the Quality of English Language

Moderate advice on English should be sought.

Author Response

Reviewer 2

The authors should consider the followings:

The authors should state whether their identified peaks, even though with HR molecular formula estimation, did the authors verify those candidate peaks with purchased or in-house chemical markers for verification? What are the false discovery rate of all the candidates concern? Did the authors use other methodology to cross check the amount and validity of all the proposed candidates? If yes, please elaborate clearly and if no, the authors should justify why not.

All the identities of the reported molecules are putative, the word identification is inappropriate, for this reason we used the term annotation. We did not perform the verification with standards because of the exploratory nature of the work and the difficulty in purchasing all the chemical species. The concept of annotation has been clearly explained in materials and methods and also the guidelines have been provided. The identity annotation has been conducted through HRMS and MS/MS structural elucidation.

How many of those candidate peaks verified by the chemical markers? And the authors should explain the ratio? 

No candidates have been verified by chemical markers

"3.8. Statistical analysis Disk diffusion results were statistically analyzed by one-way ANOVA followed by Tukey HSD multiple comparisons of means using R (v. 3.5.1) [43]. Data are presented as boxplots. Differences were considered significant for p-values < 0.05.". The authors should make a better statistical analysis plan, and include the normality test, with pipeline for both parametric and non-parametric analyses. Sample size justification should also be discussed.

The authors thank the reviewer for the suggestion and newly performed the statistical analysis, previously testing the data for normality and homogeneity of variances and then applying different parametric or non-parametric tests accordingly. Aa cited in the text, all experiments were performed in triplicate as usually accepted for this kind of analysis (Bona et al., 2016 – Journal of Applied Microbiology doi: 10.1111/jam.13282; Massa et al., 2018 – Canadian Journal of Microbiology 64647663; Bona et al., 2019 – Microbiology Research 10:83331; Clericuzio et al., 2021 – Antibiotics 10, 1424).

As for the Antibacterial assays, how did the authors verify the strain they used for testing? By pcr method?

The strain of Pseudomonas aeruginosa used for testing is a reference strain purchased from ATCC. The strain arrived with the certification of identification. 

"A filter paper disk 6 mm was placed on the colony and added with 10 μl of extracts or 10 μl of saline solution (0.8%) to evaluate the possible siderophore synthesis inhibition. The ability to produce siderophore was indicated by the occurrence of a yellow-orange halo around the colony..." The authors should explain whether the "extracts" contain interfering agents that affect the color test of the halo, which may create artifact results. Secondly, would the extract only dissolved in saline solution, or would the extract dissolved in other solvent? The authors should make sure that the solvent effect is well controlled in the experiments.

The extracts suspended in dioxane were colourless and therefore they not interfered with the detection of siderophore production. The method description contained an error: the extracts were in fact, as mentioned, resuspended in dioxane. The Authors modified the M&M section accordingly.

The authors should provide the chemical structures of all the validated or proposed chemical candidates in a figure.

A new figure has been generated and called Figure 3.

As in a supplementary figure, the authors may depict the mechanism of secondary metabolite(s) involved in the study may be illustrated. 

The Authors thank the reviewer for this suggestion. However, they feel that this work could be considered as a future development of this publication, as it is beyond the initial scope of this research.

The authors may further give details whether any of the chemical candidates undergoing clinical trials by the team or others, and the results if the trials (microbiological specific).

The authors have not used these extracts in clinical trials. Before these molecules can be tested on living beings, they still have to be tested in cell cultures.

Study limitations should be discussed in the article. 

The authors report some limitations of the work both in the Results and Discussion section and in the conclusions.The authors should clearly state the novelty of the current study in the abstract.

Thanks to the reviewer for the suggestion, the Authors add the novelty to the abstract.

Reviewer 3 Report

Comments and Suggestions for Authors

The current manuscript can be accepted for publication on condition that the authors respond to the following comments and inquiries. Upon receiving the authors’ response, the manuscript can be accepted for publication.

1.      In Table 1, a typo error was observed in exact mass.  Exact m/z should be changed to +585.1755

2.      In line 169, a typo error was observed in the LPC name. LPC name should be changed to phosphocholine.

3.      The authors should provide NMR spectra and spectral data of the following compounds in the supporting information for conformation of the structural idendity.

a.      Dehydrophlegmacin-9,10-quinone-8'-methylether (1a)  

b.      Dehydrophlegmacin-9,10-quinone (1b)

c.       phosphocholine C18 isomers

d.     oxo-octadecadienoic acid isomers

e.      Pistillarin (3)

f.        Coriolic acid (4)

g.      13-oxo-octadeca-230 9(Z),11(E)-dienoic acid (5)

Author Response

Reviewer 3

The current manuscript can be accepted for publication on condition that the authors respond to the following comments and inquiries. Upon receiving the authors’ response, the manuscript can be accepted for publication.

  1. In Table 1, a typo error was observed in exact mass.  Exact m/zshould be changed to +585.1755

Thank you for your suggestion, the error has been corrected

  1. In line 169, a typo error was observed in the LPC name. LPC name should be changed to phosphocholine.

Done

  1. The authors should provide NMR spectra and spectral data of the following compounds in the supporting information for conformation of the structural identity.
  2. Dehydrophlegmacin-9,10-quinone-8'-methylether (1a)
  3. Dehydrophlegmacin-9,10-quinone(1b)
  4. phosphocholine C18 isomers
  5. oxo-octadecadienoic acid isomers
  6. Pistillarin (3)
  7. Coriolic acid (4)
  8. 13-oxo-octadeca-230 9(Z),11(E)-dienoic acid (5)

Done

Reviewer 4 Report

Comments and Suggestions for Authors

Title:  

 A study of metabolites from Basidiomycota and their activity against Pseudomonas aeruginosa

Authors:

M. Clericuzio, G. Novello, M. Bivona, E. Gamalero; E. Bona, A. Caramaschi, N. Massa, A. Asteggiano and C. Medana

The paper focused on basidiomycota crude extracts which had proven to be active against Pseudomonas aeruginosa were subjected to liquid chromatographic separation following a fixed protocol, leading to six macro-fractions for each fungal extract. The various fractions thus obtained were tested for bioactivity against P. aeruginosa, and some of them were characterized by HPLC-HRMS and NMR.

(1) “Fractions with inhibition halo values similar or higher than those of Imipenem (A) and Meropenem (B) were considered effective against P. aeruginosa as underlined in the figure by the two blue lines.”

Please add all relative pictures of inhibition halo against P. aeruginosa.

(2) There is abnormal display on the X-axis and Y-axis in Figure 1 and Figure 2.

So it is difficulty in making judgments.

Please revise it.

Comments on the Quality of English Language

Extensive editing of English language required

Author Response

Reviewer 4

The paper focused on basidiomycota crude extracts which had proven to be active against Pseudomonas aeruginosa were subjected to liquid chromatographic separation following a fixed protocol, leading to six macro-fractions for each fungal extract. The various fractions thus obtained were tested for bioactivity against P. aeruginosa, and some of them were characterized by HPLC-HRMS and NMR.

(1) “Fractions with inhibition halo values similar or higher than those of Imipenem (A) and Meropenem (B) were considered effective against P. aeruginosa as underlined in the figure by the two blue lines.” Please add all relative pictures of inhibition halo against P. aeruginosa.

The Authors are very sorry, but unfortunately the did not documented this part with photos. 

(2) There is abnormal display on the X-axis and Y-axis in Figure 1 and Figure 2. So it is difficulty in making judgments. Please revise it.

The Authors modified Figures 1 and 2 and they hope now they are easily readable.

Extensive editing English required

The authors had the English language revised by a native speaker

Round 2

Reviewer 4 Report

Comments and Suggestions for Authors

A study of metabolites from Basidiomycota and their activity against Pseudomonas aeruginosa 

M. Clericuzio, G. Novello , M. Bivona , E. Gamalero ; E. Bona, A. Caramaschi , N. Massa 1, A. Asteggiano and C. Medana 

The manuscript can be accepted in present form.

Comments on the Quality of English Language

Minor editing of English language required.